# O-GlcNAcylation Is Required for the Survival of Cerebellar Purkinje Cells by Inhibiting ROS Generation

**DOI:** 10.3390/antiox12040806

**Published:** 2023-03-26

**Authors:** Fengjiao Liu, Shen Li, Xin Zhao, Saisai Xue, Hao Li, Guochao Yang, Ying Li, Yan Wu, Lingling Zhu, Liping Chen, Haitao Wu

**Affiliations:** 1Beijing Institute of Basic Medical Sciences, Beijing 100850, China; 2Key Laboratory of Neuroregeneration, Co-Innovation Center of Neuroregeneration, Nantong University, Nantong 226019, China; 3Chinese Institute for Brain Research, Beijing 102206, China

**Keywords:** Purkinje cells, OGT, mitochondrial damage, reactive oxygen species, cerebellar development

## Abstract

Purkinje cells (PCs), as a unique type of neurons output from the cerebellar cortex, are essential for the development and physiological function of the cerebellum. However, the intricate mechanisms underlying the maintenance of Purkinje cells are unclear. The O-GlcNAcylation (O-GlcNAc) of proteins is an emerging regulator of brain function that maintains normal development and neuronal circuity. In this study, we demonstrate that the O-GlcNAc transferase (OGT) in PCs maintains the survival of PCs. Furthermore, a loss of OGT in PCs induces severe ataxia, extensor rigidity and posture abnormalities in mice. Mechanistically, OGT regulates the survival of PCs by inhibiting the generation of intracellular reactive oxygen species (ROS). These data reveal a critical role of O-GlcNAc signaling in the survival and maintenance of cerebellar PCs.

## 1. Introduction

Purkinje cells (PCs), as the sole neurons output from the cerebellar cortex, are arguably some of the most important neurons in the vertebrate CNS, with characteristic planar fan-shaped dendrites that receive inputs from a wide range of synaptic signals [1,2]. Proper PC development is essential for normal cerebellar function [3,4]. The developmental peak of PCs occurs in the postnatal period in mice. A variety of diseases (e.g., spinocerebellar ataxia [5,6], idiopathic tremor [7,8], Huntington’s disease [9] and Autism Spectrum Disorder [10]) are accompanied by PC degeneration, which is a key factor causing disordered cerebellar function. A variety of genes have been found to be involved in the maintenance of PC physiological function [11,12], but the precise molecular mechanisms underlying the development and survival of PCs are unclear.

O-linked N-acetylglucosamine (O-GlcNAc) transferase (OGT), whose gene is localized on the X chromosome, is widely distributed in the mitochondria, cytoplasms and nuclei of cells [13]. OGT can transfer UDP-GlcNAc generated by the intracellular hexosamine biosynthesis pathway to the serine and threonine residues of targeted proteins to complete the O-GlcNAcylation of proteins [14,15]. As a rapid and reversible post-translational modification, O-GlcNAcylation is present on a variety of proteins in the central nervous system [16,17] and performs many important and diverse functions [18,19]. The deletion of OGT in neural stem cells impairs corticogenesis through endoplasmic reticulum (ER) stress and the Notch signaling pathway [20,21]. In the cerebellum, OGT in granule precursor cells mediates the cerebellar development and the progression of the Shh subgroup of medulloblastoma [22], and OGT is highly expressed in the PCs [23]. However, the role of O-GlcNAcylation in PCs is still elusive.

Here, we show that OGT deletion in PCs leads to the dendritic degeneration and cell death of PCs and, consequently, impairs cognitive and motor functions in mice. In addition, OGT-deficient PCs are detected with increased levels of autophagy and ROS in mice, effects possibly caused by mitochondrial damage. Moreover, the accumulation of ROS eventually leads to the degeneration and loss of PCs. Therefore, OGT and O-GlcNAcylation are essential for the physiological function and survival of PCs.

## 2. Materials and Methods

### 2.1. Animals

L7-Cre (JAX:004146), OgtloxP/loxP (JAX:004860) and LC3-EGFP reporter (JAX:027139) mice were maintained on a C57BL/6 background, as described previously [24,25,26]. Homozygous floxed OgtloxP/loxP or OgtloxP/Y mice were crossed with L7-Cre mice expressing Cre recombinase driven by the Pcp2 promoter specifically expressed in Purkinje cells [24]. These mice were used to generate L7-Cre: OgtloxP/Y and L7-Cre: OgtloxP/loxP mice as the conditional knockout mice (cKO), whereas their OgtloxP/loxP and OgtloxP/Y mice littermates served as wildtype controls (Ctrl). LC3-EGFP mice were provided by Jianrong Wang at Suzhou University (Jiangsu, China). Ai9 mice were generated as previously described [27]. All experiments with animals were performed in conformity with the protocols approved by the Institutional Animal Care and Use Committee of Beijing Institute of Basic Medical Sciences (approval No. SYXK 2019-0004 as of September 2020). All in vivo animal experiments were conducted in compliance with the ARRIVE 2.0 guidelines [28]. All the mice were genotyped by PCR assay using murine tail DNA. All the sequences used for mice genotyping are listed in Appendix A.

For the drug treatments, cKO mice were intraperitoneally injected with 50 µg/g of Mitochondrial division inhibitor-1 (MCE, HY-15886), 20 µg/g of ROS inhibitor Mitoquinone mesylate (Selleck, S8978) or an equal volume of vehicle control (DMSO) administered daily for 8 consecutive days from P13 to P20. The mice were sacrificed at P21. For ROS detection, the Ctrl and cKO mice were intraperitoneally injected with 10 µg/g of ROS-sensitive hydroethidine dye (hyHEt) (Selleck, S7162) at P14. The mice were sacrificed 4 h later for further analysis.

### 2.2. Nissl Staining

Nissl staining of brain slices (40 μm) was performed as previously described [27]. Briefly, the mice were sacrificed at postnatal days 0 (P0), P7, P14, P21 and P28, respectively, fixed in 4% paraformaldehyde (PFA), dehydrated with 10%, 20% and 30% sucrose solution orderly and subsequently frozen sectioned. Slices of the cerebellum were immersed in 0.5% tar violet solution (Beyotime (Shanghai, China), C0117) for 20 min. Then, the slices were rinsed in distilled water, differentiated in 95% ethanol for 1 min, dehydrated in 75% ethanol, 90% ethanol and 100% ethanol for 30 s each and sealed with neutral resin.

### 2.3. Transmission Electron Microscopy

The electron microscopy of the cerebellum samples taken from mice at postnatal days 14, 17 and 21 was performed as described previously [20]. For each group, three mice were used for transcardial perfusion with PBS (10 mM, pH 7.4), and the brain tissues were fixed in 2% formaldehyde and 2.5% glutaraldehyde in 0.1 M sodium cacodylate buffer (pH 7.4). After 12 h, the cerebellum was washed thoroughly and soaked in 0.1 M sodium dimethylarsenate buffer. The cerebellum was embedded in 4% agar and trimmed with a conventional microtome. After that, the sections were fixed in 1% osmium tetroxide/1.5% potassium ferrocyanide solution for 1 h, washed three times in distilled water, incubated in 1% uranium peroxide acetate for 1 h, washed twice in distilled water and then dehydrated with gradient alcohol (50, 70, and 90%, 10 min each time; 100%, 10 min twice). Finally, the samples were incubated with propylene oxide for 1 h and then percolated overnight in a 1:1 mixture of propylene oxide and Epon (TAAB, Aldermarston, UK). On the next day, the samples were embedded in Epon and polymerized for 48 h at 60 °C. Ultrathin sections (about 60–80 nm) were cut sagittally using a Reichert Ultracut-S microtome (Leica-Reichert, Wetzlar, Germany) and placed on copper mesh stained with lead citrate. The formation of PF–PC synapses was observed by transmission electron microscopy (Hitachi, Tokyo, Japan). Using 3% glutaraldehyde, we then sectioned the cerebellum into 70 nm slices and examined them with an H-7650 transmission electron microscope (Hitachi, Tokyo, Japan).

### 2.4. Immunofluorescent Staining

The brain samples were embedded in OCT (Tissue-tek 4583, SAKURA, Tokyo, Japan), and frozen sections were performed. The immunofluorescent staining (IF) was performed as previously described [29]. Briefly, the frozen brain sections (40 μm) were washed for 10 min with 0.3% Triton X-100/PBS (PBS, Applygen, Beijing, China) three times and then covered with 3% BSA in PBST (0.3% Triton X-100, Sigma-Aldrich, South Carolina, USA) for 1 h. The sections were then incubated with primary antibodies for 12 h at 4 °C. Subsequently, the sections were washed three times for 10 min with 0.3% PBST, followed by Alexa Fluor 568- or Alexa Fluor 488-conjugated fluorescent secondary antibody incubation. Nuclear staining was directly applied using a mounting medium with DAPI. All images were processed and analyzed using Imaris (v9.6 Bitplane, Oxford Instruments Group, Concord, MA, USA), Image J software (NIH, US) and FV10-ASW (v1200, Olympus, Tokyo, Japan).

### 2.5. Golgi Staining

The FD Rapid GolgiStainTM Kit (PK401, FDNeuroTechnologies, INC., Columbia, USA) was used for Golgi staining. Three Ctrl and three cKO mice were sacrificed at P14, P17 and P21, and the brain tissues were quickly removed. The tissues were immersed in the mixed solution (A:B = 1:1) and then placed in a dark place at room temperature for 3 weeks. Then, the tissues were transferred into solution C in a dark room at room temperature for 3 days. The tissues were cut into 100 µm sections, and each section was dried at room temperature. The sections were washed in double-distilled water (ddH_2_O) twice for 3–4 min each and then placed in a mixture (D:E:ddH_2_O = 1:1:2) for 10 min. The sections were immersed in 50, 75, 95 and 100% ethanol for 4 min each. Lastly, the sections were cleared in xylene three times for 4 min each and, finally, sealed with neutral resin.

### 2.6. SFV Virus Infects Neurons In Vivo

SFV (Semliki forest virus) was generously provided by Fan jia at Shenzhen Institute of Advanced Technology, Chinese Academy of Sciences (Shenzhen, China) [30]. C57BL/6 mice were used for the viral injection. Mice were anesthetized by injection with 1% sodium pentobarbital (50 mg/kg) and placed in a stereotaxic apparatus. Carefully, we cut the skin of the mouse and drilled the skull. The SFV virus (100 nL) was injected into the brain through a 10 μL syringe. After 24 h, the mice were sacrificed and transcardially perfused with 0.9% saline followed by 4% PFA. The brains were removed and post-fixed overnight in 4% paraformaldehyde and cut into 40 μm sections. The brain sections were stained with DAPI and imaged using a confocal microscope (Olympus, Tokyo, Japan).

### 2.7. Real-Time Quantitative RT-PCR

A total of 50 mg of cerebellum tissue samples from the Ctrl and cKO mice was homogenized with 1 mL of Trizol reagent (Invitrogen, Carlsbad, USA) in a glass homogenizer on ice. Then, the mixture was left at RT for 10 min. A total of 200 microliters (200 μL) of chloroform was added and the tubes were vortexed for 1 min and then centrifuged at 15,000 × *g* for 15 min at 4 °C. The supernatant was transferred into a new centrifuge tube, mixed with 500 μL of isopropanol, vortexed for 1 min and then centrifuged at 15,000× *g* for 10 min at 4 °C. The precipitant was washed twice using 70% ethanol, dried and dissolved in RNase-free water (0.1%). The total RNA concentration was measured using a Biophotometer Plus spectrometer (Eppendorf, Hamburg, Germany). The total RNA (1 μg) was reverse-transcribed into cDNA using HiScript Reverse Transcriptase (Vazyme, Nanjing, China). The cycling conditions used to acquire the cDNA were 25 °C for 5 min, 50 °C for 45 min and 85 °C for 2 min. The mRNA of the target genes was detected by using an HiScript II One Step qRT-PCR SYBR Green Kit (Vazyme, Nanjing, China). The cycling conditions for qPCR were 95 °C for 2 min, 95 °C for 10 s, 60 °C for 20 s, 95 °C for 15 s, 60 °C for 60 s and 95 °C for 15 s. The cycling number was 35. A quantitative PCR assay was carried out using an MX3005P real-time PCR thermocycler (Roche, Porterville, CA, USA). The PCR primers used are listed in Appendix A. Additionally, the relative mRNA level was calculated according to the Ct value.

### 2.8. Accelerating Rotarod

The mice were first adapted to the stick and then measured every 8 h on 8 consecutive occasions. In each test, the speed was accelerated from 4 rpm to 60 rpm every 5 min with a limit of 300 s.

### 2.9. Behavior in the Three-Chamber Social Test

A 41 cm (wide)× 70 cm (length)× 28 cm (high) three-chambered apparatus was constructed, and the video was taped from the frontal perspective of the apparatus. The video recordings were collected from above. The bottom panel was white, while all other walls were blue. Next, social reorganization and social memory preference assessments were conducted. During a 10-minute habituation period, the subject mice were placed in the middle chamber with all the doors open, and each outside chamber contained one empty black wire cup. In the social approach stage, the subject mice were placed in the middle chamber with the doors closed, while an unfamiliar C57BL/6 mouse was placed in one cup (S1), and the other cup was empty (E). The duration of time spent in the S1 or E chamber was measured in a 10-minute session, in which the subject mice were required to be at a social distance from the cup. In the social novelty stage, the subject mouse was placed in the middle chamber with the doors closed, while the new unfamiliar C57BL/6 mouse was placed in the empty cup (S2). The duration of time spent in the S1 or S2 chamber was measured for 10 min, and the subject mice were required to be at a social distance from the cup.

### 2.10. Elevated Plus Maze Test

A 40 cm (length) × 10 cm (width)× 50 cm (height) elevated plus maze was used to measure anxiety-like behavior. The closed arms were enclosed by 20 cm-high walls. The subject mice were placed in the center facing one of the open arms when the test started, and the duration of time spent in each arm was measured for 10 min.

### 2.11. Catwalk

The catwalk test was performed in a dark place, and the subject animal was placed in the same area. Each subject mouse traversed from left to right, and the footprints on the runway were recorded. A footprint was scored as valid if the mouse passed through the channel normally, whereas a footprint was scored as invalid if the mouse stopped halfway or dropped off the catwalk.

### 2.12. Quantitative-Tandem-Mass-Tag-Based Proteomic Analysis

The total protein from the cerebellum samples of the Ctrl and L7 cKO mice was extracted at P14. The concentration of the extracted protein sample was measured using a BCA protein assay kit (Thermo, 23225, MA, USA), and the protein quality was confirmed using SDS-PAGE. Then, the extracted protein sample was digested with trypsin and labeled with tandem mass tags. Additionally, an equal amount of each labeled sample was mixed for chromatographic separation and analyzed using LC-MS/MS. After the quality assessment and preprocessing, expression level analysis and function analysis of the proteins were conducted, respectively. A KEGG signaling pathway enrichment analysis was performed to analyze the relevant signaling pathways of the differentially expressed proteins, and the heat map profiling of the expression levels of autophagy and ROS generation in the Ctrl and cKO mice was conducted.

The LC-MS data were deposited in the ProteomeXchange Consortium via the iProX partner repository (dataset identifier PXD040011/IPX0005887000). The entire datasets are publicly available at https://www.iprox.cn//page/project.html?id=IPX0005887000 (accessed on 2 October 2022).

### 2.13. Detection of Endogenous ROS Levels 

The detection of endogenous ROS level in PCs (IF) was performed as previously described [31]. In vivo ROS levels were determined using the ROS-sensitive dye, hudroethidine. The mice at P14 or P17 were intraperitoneally injected with hudroethidine (10 µg/g) 4 h before the mice were sacrificed by injection with 1% sodium pentobarbital (80 mg/kg). Then, the mice were used for transcardial perfusion with paraformaldehyde (PFA) and PBS (10 mM, pH 7.4), and the brain tissues were fixed in 4% paraformaldehyde (PFA), dehydrated with 10%, 20% and 30% sucrose solution orderly and subsequently frozen sectioned. The brain sections were mounted in mounting media containing DAPI and imaged using a confocal microscope (Olympus, Tokyo, Japan). 

### 2.14. Statistical Analysis

All experiments were repeated at least three separate times. The data compared between the different groups were shown as the mean ± SEM. For statistical analysis, *p* < 0.05 (*), *p* < 0.01 (**), *p* < 0.001 (***), *p* < 0.0001 (****) or no significance (n.s.) were determined by the unpaired t-test, one-way or two-way ANOVA analysis and the multiple comparisons test using GraphPad Prism 9 and SPSS. All images were analyzed using Image J (NIH, US) and NDP.View2 software (Hamamatsu Photonics, Shizuoka, Japan). 

## 3. Results

### 3.1. Loss of OGT in PCs Causes Cell Loss and Cerebellar Atrophy

O-GlcNAcylation is critical for the development and function of the brain within the nervous system. Additionally, OGT is abundant in PCs [23]. The peak period of PC development occurs after birth in mice, and there is a phasic change in PC development [32,33]. PCs in newborn mice exhibit a multilayered distribution and begin to exhibit a monolayer distribution by around P5, and the dendrites of the PCs grow rapidly from P7 to P14 [33]. The key stage in which the dendritic structure of PCs establishes stable and proper synaptic connections with multiple nerve fibers is the period from P14 to P21. Additionally, the dendritic development of PCs is largely completed by P21 [33,34]. Therefore, we selected mice at P7, P14 and P21 to examine the O-GlcNAcylation levels of PCs in the physiological state and found that the cells in the PC layer (PL) exhibited a sustained intracellular high O-GlcNAcylation (Appendix A). Because there is a high level of intracellular O-GlcNAcylation in PCs, the role of O-GlcNAcylation in PCs merits further exploration.

To investigate the role of O-GlcNAcylation in PCs, we used the L7-Cre driver to selectively delete Ogt in the PCs. L7-Cre is exclusively expressed in PCs in the cerebellum, consistent with our results on reporter mice utilizing the genotype L7-Cre: Ai9 (Appendix A). L7 cKO mice (L7-Cre−/+: OgtloxP/Y and L7-Cre−/+: OgtloxP/loxP) were born at a normal rate and developed similarly to their wildtype littermates (OgtloxP/Y and OgtloxP/loxP), which served as the control (Ctrl) group. First, we confirmed that the high O-GlcNAcylation level was abolished in the PCs of the cerebellum in the cKO mice at P14, P17 and P21 (Appendix A). Additionally, the abolishment of O-GlcNAcylation did not affect the lifespan of the cKO mice compared to the Ctrl. We then explored the macroscopic appearance of morphological changes in the cerebellum of Ctrl and cKO mice. The Nissl staining results showed that the cerebellum of cKO mice did not display obvious morphological changes until P14. However, the cerebellum began to shrink from P14 (Figure 1A). Additionally, the area of the sagittal section of the cerebellum gradually decreased from P14 (Figure 1B). Meanwhile, the immunofluorescence staining results based on calbindin, a marker for PCs, for the cerebellum sagittal sections showed that the PCs began to disappear in the cKO mice from P14 (Figure 1C,D). In addition, the key stage of PC loss occurred from P14 to P21 (Figure 1C,D). Moreover, the thickness of the molecular layer significantly decreased from P14 to P21 (Figure 1E). This period is also marked the beginning of cerebellar atrophy. In conclusion, the deficiency of O-GlcNAc in PCs results in cell loss, and peak of PC loss occurs from P14 to P21, accompanied by cerebellar atrophy.

### 3.2. Mice Lacking O-GlcNAc in PCs Exhibit Locomotor and Social Memory Dysfunctions

The cerebellum is involved in a variety of motor and cognitive functions [35]. To assess the behavioral consequences of O-GlcNAcylation deficiency in PCs, we conducted assays aiming to test motor and nonmotor functions of the cKO mice. A catwalk test was used to analyze the gait of the mice [36]. The results showed that the gait of the cKO mice changed significantly. The BOS (base of support) of the front and hind paws of cKO mice increased significantly, the stride length of each step decreased significantly, and the swing time of the paws in the air was significantly shorter (Figure 2A–E). Therefore, the cKO mice exhibited a minced gait. Meanwhile, the results of the rotarod analysis showed that the residence time of the cKO mice on the rolling bar was shorter than that of their wildtype littermates (Figure 2F,G). In addition, the cKO mice preferred to stay in the open arm in the elevated cross maze experiment (Figure 2H,I). Furthermore, in the three-box social experiment, the cKO mice could clearly identify empty cages and unfamiliar mice but could not distinguish unfamiliar mice from familiar mice (Figure 2J–L). Therefore, the cKO mice had social memory deficits. In conclusion, the mature mice with OGT-deficient PCs showed behavioral dysfunction and social memory impairment.

### 3.3. Dendrites and Spines of PCs Were Severely Degenerated in cKO Mice

The results outlined above indicate that the critical period of PC cell degeneration in cKO mice is from P14 to P21. Therefore, we wondered whether PCs undergo neural alterations in this specific stage. First, we characterized the morphology of the PCs from the cKO mice at P17 using sparse-virus Semliki forest virus (SFV) labeling. Additionally, 3D stereo reconstruction of the PCs was performed using Imaris software. The results showed that the dendrites and spines of the PCs from the cKO mice appeared to be significantly degenerated at P17 (Figure 3A). The PCs showed significant reductions in the total dendritic length, average diameter of the dendrites, spatial volume occupied by dendrites, density of the dendritic spines, and cell soma volume (Figure 3B–F). Interestingly, cKO mice had a significantly higher number of primary dendrites in their PCs (Figure 3G), suggesting an impairment of cell polarity in the PCs. To further explore the morphological changes during PCs degeneration, we reconstituted and counted morphological changes in the PCs using Golgi staining at P14, P17 and P21 (Figure 3H). Firstly, the total length of the dendrites and the average diameter of dendrites decreased from the early to the terminal stages of PC degeneration (Figure 3I,J). Secondly, the degeneration of spines did not occur at the beginning, and reductions in the spin density appeared only in the middle and late stages of PC degeneration (Figure 3K). However, the degeneration of dendrites appeared to occur simultaneously from the distal and proximal ends of the cell soma (Figure 3L). Moreover, we observed morphological changes in the axons of OGT-deficient PCs. We found that the axons in OGT-deficient PCs showed discontinuous intumescence in axons with a large number of nodules, and the number of axons dramatically decreased (Appendix A). Therefore, the depletion of O-GlcNAcylation in PCs can lead to dramatical morphological degenerations, especially in the case of axons, dendrites and dendritic spines.

### 3.4. Upregulated Mitochondrial Damage, ROS Levels and Autophagy in OGT cKO Mice

To explore the molecular mechanism underlying the degeneration and death of PCs caused by their loss of OGT, a quantitative proteomic analysis of proteins from the Ctrl and cKO mice was performed at P14. The results of the omics analysis of cerebellar protein content using tandem mass tags (TMT) for the P14 mice demonstrated that 155 proteins were significantly changed in the cKO mice (63 up-regulated proteins vs. 92 down-regulated proteins: absolute Log2fold-change > 0.2 and FDR < 0.05) compared with the Ctrl mice (Figure 4A). The PC marker, calbindin, was also significantly decreased, which indicated that the data were credible. Function enrichment analysis (KEGG enrichment) of these differentially expressed proteins revealed enrichment in several pathways, such as the PI3K-Akt, calcium, autophagy and Ras signaling pathways (Figure 4B). The PI3K-Akt, calcium and lysosome signaling pathways are related to the autophagy and production of reactive oxygen species (ROS) [37,38,39,40]. In the autophagy and ROS signaling pathways, the expression of vital proteins such as Itpr1, Prkcd, Prkcd, Mtmr3, Pdk1 and Nos1 was significantly down-regulated in the cKO mice (Figure 4C,D). Meanwhile, we performed quantitative PCR to analyze the expression of proteins in the autophagy signaling pathway using the Ctrl and cKO cerebellum samples at P14. The mRNA levels of Atg4b, Atg6L1-1, Atg6L1-2, Atg6L1-3 and Atg7 were significantly up-regulated (Figure 4E). In the ROS generation signaling pathway, the genes clearing the ROS, such as Sod2, CAT and GPX, were significantly decreased. On the contrary, the genes producing ROS were clearly up-regulated, including NOX2, MPO and NOS1 (Figure 4F). These results indicate autophagy and ROS generation were activated in the cerebella of the cKO mice. Furthermore, the cerebella of the cKO mice also presented changes in the expression of genes involved in basic mitochondrial biological processes and mitophagy, such as Nrf1-5, Ppargc1b, Pink1 and ULK2 (Figure 4G,H). These results further suggest that there was clear mitochondrial damage to the PCs of the cKO mice in the initial stage of degeneration. Damaged mitochondria often cause the excessive intracellular release of reactive oxygen species (ROS) and their accumulation [20,41]. The overproduction of mitochondrial ROS, which causes neurodegeneration and cell death, is closely associated with the majority of neurological conditions and neurodegenerative diseases [42,43]. Hence, we hypothesized that OGT might mediate the degeneration and death of PCs by regulating mitochondrial function and ROS signaling.

To further clarify the subtle changes in the PCs after OGT knockout, the subcellular structure of PCs from the Ctrl and cKO mice was examined using electron microscopy. The electron microscopy results showed that there were a large number of autophagosomes in the PCs of the cerebella of the P14, P17 and P21 cKO mice compared with the Ctrl mice (Figure 5A). In addition, autophagosomes encapsulating mitochondrial structures and damaged mitochondria were observed in the OGT-deficient-PCs (Figure 5A and Appendix A). In order to further confirm that OGT deficiency in the PCs can lead to mitochondrial damage, we used conventional methods to analyze the changes in mitochondrial mass in the OGT-deficient PCs and found that mitochondrial mass was significantly reduced in the OGT-deficient PCs based on quantitative immunostaining for translocase of the outer membrane 20 (TOMM20) (Appendix A). This result is consistent with the results obtained using electron microscopy. Meanwhile, we found a high accumulation of ROS in the PC layer in the cKO mice (Figure 5B,E). This result suggests that mitochondrial damage in the early stage of PC degeneration may lead to high intracellular ROS accumulation.

LC3 is a classical marker protein of autophagosomes [44]. To further verify that autophagy exists in PCs, IF staining was used to detect the expression of LC3. An elevation of LC3 expression occurred in PCs of the cKO mice at P14, with the notable appearance of LC3 aggregates generated at P17 (Figure 5C,F). Additionally, LC3-GFP gene reporter mice were utilized to similarly verify the generation of LC3 aggregates in the PCs. The results showed that the great enrichment of LC3-GFP disappeared in cKO mice containing the LC3-GFP reporter at P14 and P17, compared with the Ctrl mice (Figure 5D,G). P62 can mediate the process of degrading waste in cells, mark damaged mitochondria through the PINK1/Parkin signaling pathway, and transport them to the interior of autophagosomes for degradation [45,46]. Beclin1 acts as one of the most important protein complexes in the formation of the autophagy pathway, being in action from the formation of the autophagosome to its elongation and maturation [47]. The immunofluorescence results showed that the protein levels of Beclin1 and P62 were increased and reduced, respectively, in the OGT-deficient PCs (Appendix A). Together, these results demonstrate that the PCs underwent a significant activation of autophagy and ROS during degeneration.

### 3.5. Mitochondrial Damage and ROS Production Lead To Degeneration in OGT-Deficient PCs

To investigate the involvement of mitochondrial function and the generation of ROS in the degeneration and death of PCs in the cKO mice, we treated the Ctrl and cKO mice with a mitochondrial function protector, Mitochondrial division inhibitor-1 (Mdivi-1), and ROS inhibitor Mitoquinone mesylate (MitoQ10), administered intraperitoneally, and observed the number and morphology of PCs from these mice. Mdivi-1 and MitoQ10 have previously been shown to modulate mitochondrial function and inhibit ROS generation in the brain following their systemic delivery, respectively [48,49,50]. The Mdivi-1 treatment increased the number of PCs and the area of the cerebella of the cKO mice (Figure 6A–C). Meanwhile, the number of PCs and the area of the cerebella of the cKO mice were also increased in the cKO mice treated with MitoQ10, compared to the DMSO group (Figure 6D–F). Additionally, the ROS production in the cKO mice was significantly inhibited by the MitoQ10 treatment (Appendix A). These results indicate that the protection of mitochondrial function and inhibition of ROS generation alleviates the loss of PCs and cerebellar dysplasia. Together, these data suggest that OGT promotes the survival of PCs by mediating mitochondrial function and inhibiting ROS generation.

## 4. Discussion

In this study, we demonstrated that OGT plays a vital role in the maintenance and survival of PCs. The PCs without O-GlcNAcylation degenerated rapidly during the short development period of the dendrites from P14 to P21, which was mainly manifested in the significant degradation of dendrites and spines. Additionally, the loss of OGT and O-GlcNAc induced mitochondrial damage and the generation of ROS, which led to the degeneration and death of the PCs.

The development and survival of Purkinje cells, as the only output of the cerebellum, affects cerebellar morphology and function [51]. Hence, the crosstalk between molecules involved in regulating PC development and survival, such as neurotrophins and insulin-like growth factor, is essential for ensuring a fully developed and functional cerebellum [52,53]. Our previous results implicated the receptor for activated C kinase 1 (Rack1) controlling parallel fiber-Purkinje cell synaptogenesis and synaptic transmission in PCs [54]. Here, we identified O-GlcNAc as a novel post-translational modification mechanism that controls the development and survival of PCs. O-GlcNAc is a nutrient sensor. Nutrient availability is essential for the regulation of cell survival. We proved that in the cerebellum, OGT and O-GlcNAcylation play critical roles in granule neuron precursor (GNP) proliferation during cerebellar development [22]. Therefore, O-GlcNAcylation may have other important regulatory effects on the development of other cerebellar cells and interactions between different cells, which need to be explored further. 

OGT broadly affects cellular functioning through the effects of O-GlcNAc on the serine and threonine residues of over 1000 different intracellular proteins [55]. OGT regulates the proliferation of GNP through the O-GlcNAcylation of the serine 355 of Gli2 in the developing cerebellum in mice [22]. In the intestine, hypodermal cells, neurons from *Caenorhabditis elegans* and the mammalian Hela cells, OGT mediates the O-GlcNAcylation of snap29 and regulates autophagy in a nutrient-dependent manner [56]. The deletion of OGT in hematopoietic stem cells (HSCs) accumulates defective mitochondria and destroys mitophagy through decreasing the key mitophagy regulator, Pink1 [57]. However, the loss of OGT triggers the decrease of mitochondrial mass and the up-regulation of autophagy in the PCs in our study. The difference in the change in mitochondrial mass and formation of autophagy caused by OGT deletion in HSCs and PCs is probably due to the fact that HSCs are pluripotent stem cells, and PCs are committed cells. In the neural stem cells (NSCs), OGT regulates the survival of cortical NSCs and adult hippocampal NSCs through the Notch signaling pathway and STAT3, respectively [21,58]. Therefore, the centers of energy supply and O-GlcNAcylation-modified target proteins may be different between HSCs and PCs, which might contribute to the phenomenon that the loss of OGT has inconsistent phenotypes on mitochondrial mass and autophagy between HSCs and PCs. In this study, we found that the deletion of OGT induces autophagy and ROS generation. However, the key target molecules involved in autophagy and ROS generation in depleted PCs have still not been identified. Therefore, the precise molecular mechanism determining how OGT mediates autophagy and ROS generation in PCs should be elucidated in future studies. 

The death of neurons is a systematic and complex process that occurs by many different means in different situations, such as apoptosis, autophagy, ferroptosis and necrosis [59]. In this study, mitochondrial damage and the generation of ROS were found to be the main factors inducing the degeneration and death of OGT-deficient PCs. However, significant cleaved caspase3 activation was observed in the PCs throughout the degeneration period (Appendix A). These data suggest that apoptosis may be one of the causes involved in Purkinje cell degeneration and death. Therefore, extensive mitochondrial damage and ROS generation may not be the only reasons for Purkinje cells death but rather the major causes. On the other hand, mitochondrial damage and the accumulation of ROS are important intracellular factors that contribute to triggering apoptosis [60,61]. Therefore, the activation of caspase3 in OGT-deficient PCs may be induced by hyperactivated mitophagy and the overproduction of ROS. Additionally, mitochondrial damage and ROS up-regulation in OGT-deficient Purkinje cells are accompanied by increased autophagy, whose role in PC maintenance remains unclear.

Granule neurons are the most numerous neurons in the cerebellum [62]. In this study, cerebellum developmental defects were followed by the degeneration and death of PCs. To elucidate the role of O-GlcNAcylation in PCs in regard to granule neurons, the production and death of granule neurons were analyzed. The thickness of the outer granule layer during the peak period of granule neuron proliferation was not significantly altered (Appendix A). However, a notable apoptotic activation of the inner granule neuron layer after the onset of PC degeneration was identified. These results suggest that the degeneration of PCs causes granule neuron death in a non-cell autonomous manner. These results illustrate that the degeneration of PCs causes the death of granule neuron in the cerebellum both during the developmental period of PCs and in their functional maintenance, as well as the important role of PCs in the cerebellum.

## 5. Conclusions

In summary, this study revealed that the O-GlcNAcylation mediated by OGT is pivotal for the maintenance and survival of PCs. The deletion of OGT in PCs leads to the degeneration of PCs and cerebellar malformations. Meanwhile, the behavioral phenotypes observed in OGT loss-of-function mice demonstrate impaired locomotor coordination and sociability. OGT protects the PCs from mitochondrial damage and the generation of ROS, which induce the death of PCs. These results suggest that OGT and O-GlcNAcylation could potentially serve as candidate molecular targets for the treatment of neurodegenerative diseases associated with Purkinje cell loss.

## Figures and Tables

**Figure 1 antioxidants-12-00806-f001:**
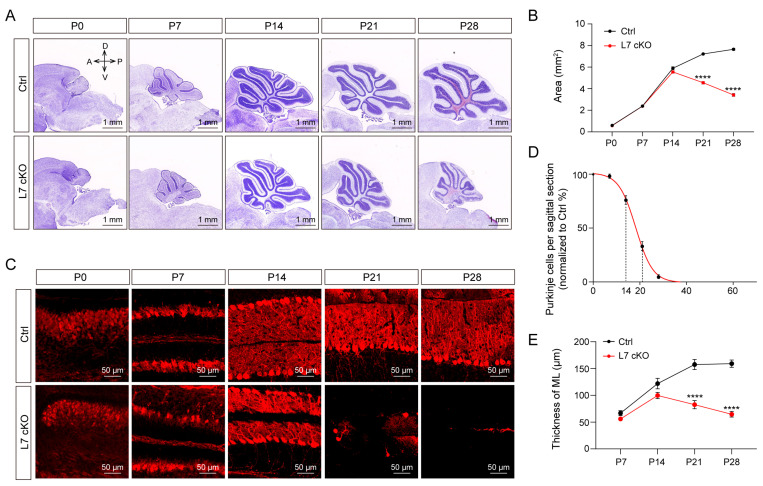
Loss of OGT in PCs induces cerebellar atrophy and PCs loss. (**A**) Nissl staining of sagittal histological sections of the cerebellar vermis showing cerebellar atrophy (scale bar: 1 mm). (**B**) Quantification of the area of the sagittal sections of the cerebellar vermis for the Ctrl and cKO mice (mean ± SEM; **** *p* < 0.0001, n > 3). (**C**) Immunofluorescent staining with anti-calbindin antibody (red) to reveal PCs in the cerebellar vermis (scale bar: 50 μm). (**D**) PCs per sagittal histological section of the cerebellar vermis is expressed as a percentage for the Ctrl mice. (**E**) The ML thickness of the cKO and Ctrl mice (mean ± SEM; **** *p* < 0.0001, n > 3). a, anterior; d, dorsal; p, posterior; v, ventral; ML, molecular layer.

**Figure 2 antioxidants-12-00806-f002:**
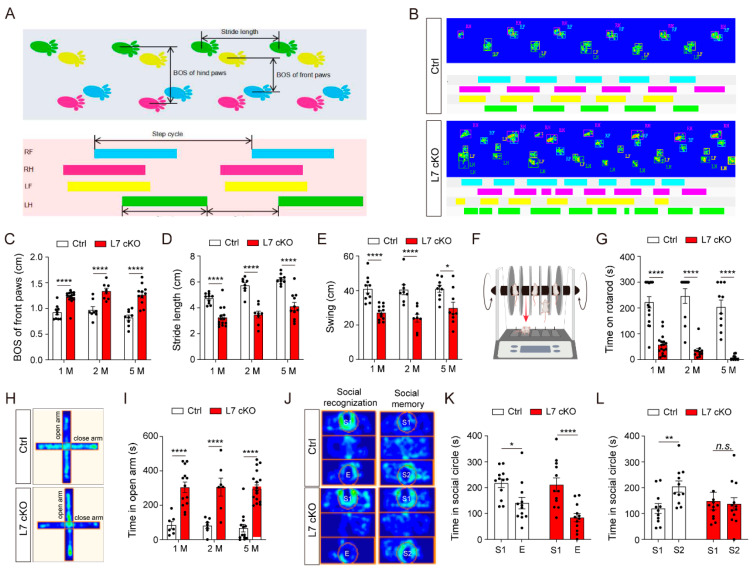
PC-specific OGT null mice show impaired motor coordination and social memory. (**A**,**B**) Representative catwalk footprints of Ctrl and cKO mice. (**C**,**D**) Quantification of the length of BOS of the front and hind paws (mean ± SEM; **** *p* < 0.0001, n > 7). (**E**) Quantification of the swing time. Only the results for the right front paw (RF) are shown (mean ± SEM; * *p* < 0.05, **** *p* < 0.0001, n > 7). (**F**,**G**) Time spent on the accelerating rotarod by the Ctrl and cKO mice (mean ± SEM; **** *p* < 0.0001, n > 7). (**H**) Heat map representing the positions of the mice in the open field. (**I**) Time spent in the open arms by the Ctrl and cKO mice (mean ± SEM; **** *p* < 0.0001, n > 7). (**J**) Heat map representing the positions of the test mice. (**K**,**L**) Time spent in the social areas of the empty (E), stranger 1 (S1) or stranger 2 (S2) cages (mean ± SEM; * *p* < 0.05, ** *p* < 0.01, **** *p* < 0.0001, n.s. = no significance, n > 10). RF, right front paw; RH, right hind paw; LF, left front paw; LH, left hind paw.

**Figure 3 antioxidants-12-00806-f003:**
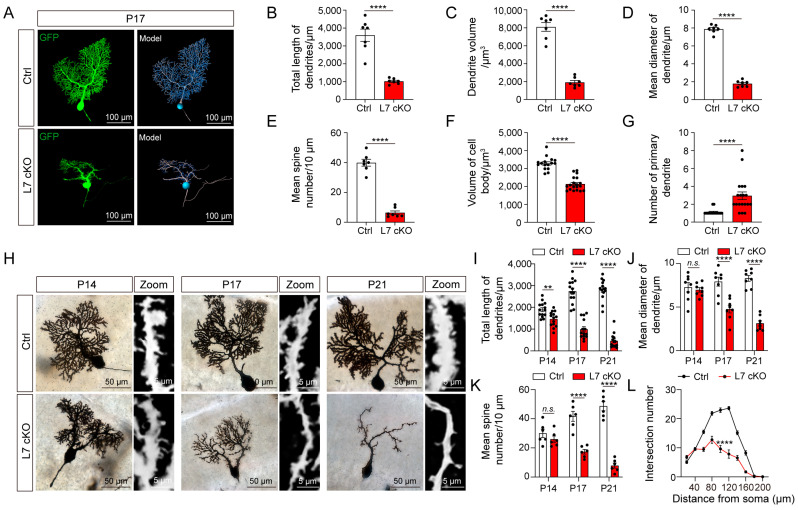
PCs were degenerated in the cKO mice. (**A**) The cerebella of the Ctrl and cKO mice were infected with a virus, and infected positive PCs exhibited by GFP (green) and the reconstruction of PCs using Imaris are shown (scale bar: 100 μm). (B-G) Morphometric analyses of the total length of dendrites (**B**), dendrite volume (**C**), dendrite mean diameter (**D**), spine density (**E**), volume of the cell body (**F**) and number of primary dendrites (**G**) of PCs (mean ± SEM; **** *p* < 0.0001, n > 5). (**H**) The morphology of the PCs after Golgi impregnation (scale bar: 50 μm). A zoom showing a high-magnification picture of the PC dendrites (scale bar: 5 μm). (**I**–**K**) Quantification of the total length of the dendrites (**I**), dendrite mean diameter (**J**) and spine density (**K**) (mean ± SEM; ** *p* < 0.01, **** *p* < 0.0001, n.s. = no significance, n > 5). (**L**) Sholl analysis of the intersection number of the dendrites at different distances from the cell body of the PCs in the Ctrl and cKO mice (mean ± SEM; **** *p* < 0.0001, n > 5).

**Figure 4 antioxidants-12-00806-f004:**
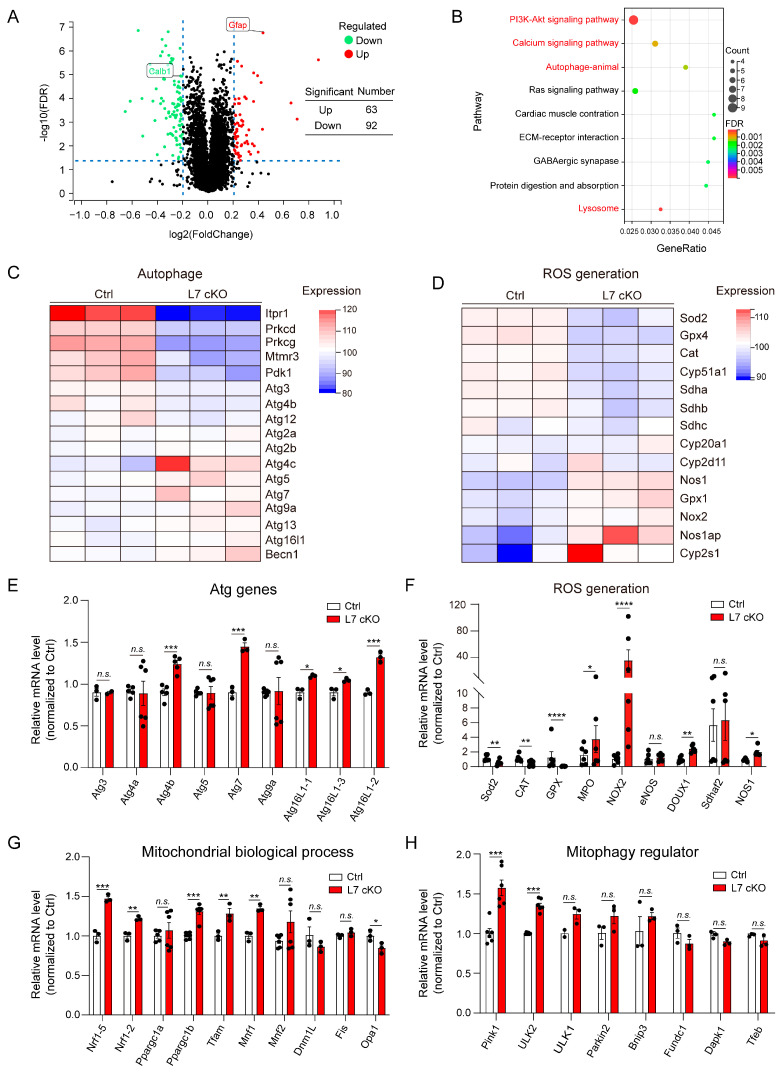
Ablation of OGT in PCs induces changes in the gene expression of mitochondrial autophagy and ROS accumulation in cKO mice. (**A**) Volcano map of differentially expressed proteins analyzed using TMT in the cerebellum of the Ctrl and cKO mice. (**B**) Curated KEGG pathway terms of up- or down-regulated proteins in the cerebellum of the cKO mice compared with Ctrl mice. The pathways closely related to autophagy are marked in red. (**C**,**D**) Heatmap of the expression of proteins in autophagy (**C**) and ROS generation (**D**) signaling pathways. The mean expression levels in the heatmap were standardized by taking the z-score within each row (n = 3). The top bar indicates the genotypes of the samples. (**E**,**F**) Quantitative RT-PCR analysis indicates significant changes in Atg-related genes (**E**) and ROS generation (**F**) in the cerebellum of the Ctrl and cKO mice (mean ± SEM; * *p* < 0.05, ** *p* < 0.01, *** *p* < 0.001, **** *p* < 0.0001, n.s. = no significance, n > 3). (**G**,**H**) Quantitative RT-PCR analysis indicates significant changes in mitochondrial biological processes-related genes (**G**) and mitophagy-related genes (**H**) in the cerebellum of the Ctrl and cKO mice (mean ± SEM; * *p* < 0.05, ** *p* < 0.01, *** *p* < 0.001, **** *p* < 0.0001, n.s. = no significance, n > 3).

**Figure 5 antioxidants-12-00806-f005:**
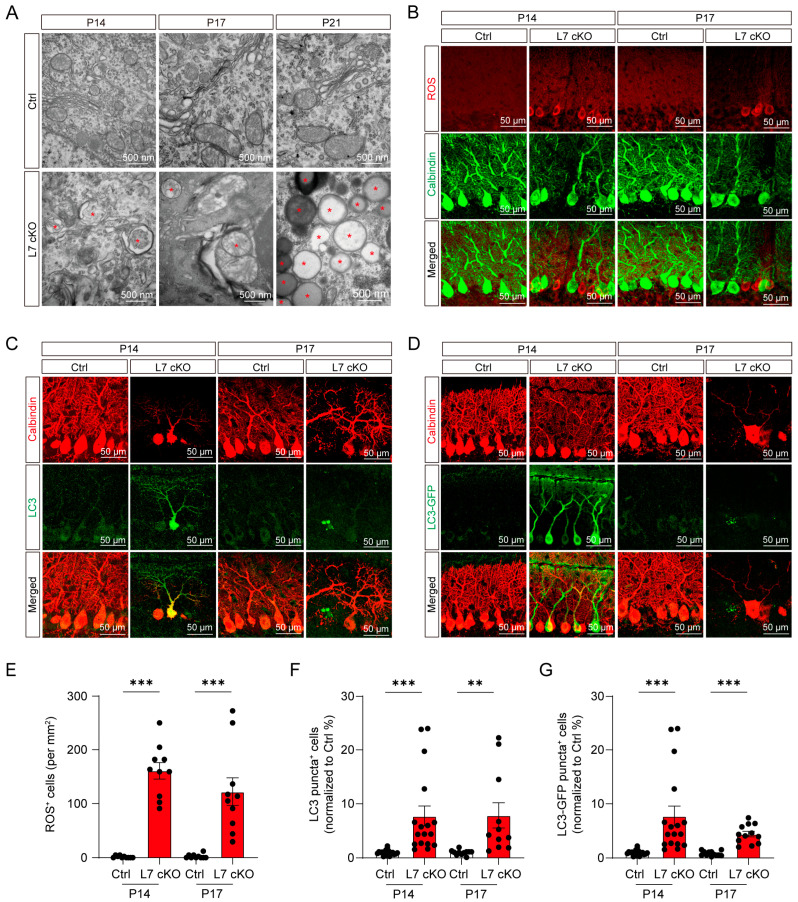
Mitochondrial damage with ROS and LC3 accumulation in OGT-deficient PCs. (**A**) Electron micrographs of PCs from the Ctrl and cKO mice at P14, P17 and P21, respectively. Autophagosome-like structures (labeled with asterisks) in PCs are labeled (scale bar: 500 nm). (**B**) The fluorescence plots show that ROS (red) were significantly elevated in the PL (scale bar: 50 μm). (**C**) Co-immunofluorescent staining with anti-LC3 and anti-calbindin antibodies showing the significantly increased LC3 expression (scale bar: 50 µm). (**D**) Immunofluorescence analysis of the cellular localization of LC3 (green) and immunofluorescent staining with anti-calbindin antibody (red) (scale bar: 50 μm). (**E**) Quantification of the ROS fluorescence intensity of the PL from the Ctrl and cKO mice (mean ± SEM; *** *p* < 0.001, n = 10). (**F**) Quantification of the LC3 puncta-positive cells from the Ctrl and cKO mice at P14 and P17 (mean ± SEM; *** *p* < 0.001, ** *p* < 0.01, n > 10). (**G**) Quantification of LC3 puncta-positive cells among PCs from the Ctrl and cKO mice at P14 and P17 (mean ± SEM; *** *p* < 0.001, n > 10).

**Figure 6 antioxidants-12-00806-f006:**
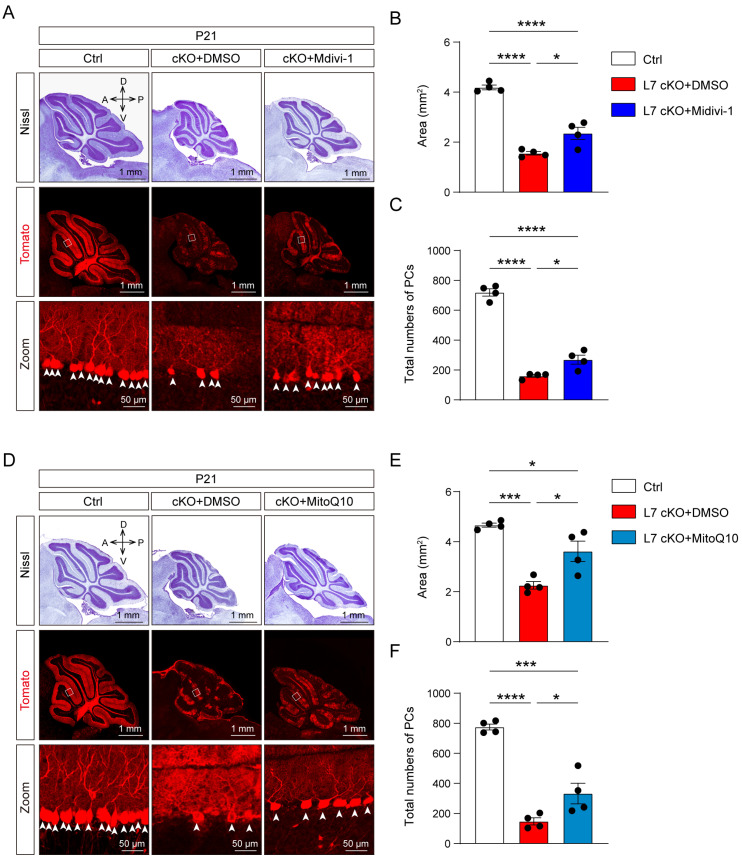
Suppression of ROS production significantly rescue cerebellar PC loss in cKO mice. (**A**) Nissl staining of sagittal histological sections of the cerebellar vermis and immunofluorescent image of Tomato (red) staining to reveal PCs in the cerebellar vermis of the Ctrl and cKO mice treated with DMSO or the cKO mice treated with Midiv-10 (scale bar: 1 mm, 50 μm). (**B**) Quantification of the area of sagittal sections of the cerebellar vermis from the Ctrl and cKO mice (mean ± SEM; **** *p* < 0.0001, * *p* < 0.05, n = 4). (**C**) The number of PCs per sagittal histological section of the cerebellar vermis from the Ctrl and cKO mice (mean ± SEM; **** *p* < 0.0001, * *p* < 0.05, n = 4). (**D**) Nissl staining of sagittal histological sections of the cerebellar vermis and immunofluorescent image of Tomato (red) staining to reveal PCs in the cerebellar vermis of the Ctrl and cKO mice treated with DMSO or the cKO mice treated with MitoQ10 (scale bar: 1 mm, 50 μm). (**E**) Quantification of the area of sagittal sections of the cerebellar vermis of the Ctrl and cKO mice (mean ± SEM; *** *p* < 0.001, * *p* < 0.05, n = 4). (**F**) The number of PCs per sagittal histological sections of the cerebellar vermis of the Ctrl and cKO mice. (mean ± SEM; **** *p* < 0.0001, *** *p* < 0.001, * *p* < 0.05, n = 4). a, anterior; d, dorsal; *p*, posterior; v, ventral.

## Data Availability

LC-MS data reported in this paper have been deposited to the ProteomeXchange Consortium via the iProX partner repository (dataset identifier PXD040011/IPX0005887000). The entire datasets are publicly available at https://www.iprox.cn//page/project.html?id=IPX0005887000 (accessed on 14 August 2022).

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
