# Peer review of "O-GlcNAcylation Is Required for the Survival of Cerebellar Purkinje Cells by Inhibiting ROS Generation"

_antioxidants, 2023, doi:10.3390/antiox12040806_

Round 1
Reviewer 1 Report
The authors of this manuscript report an important function of O-GlcNAcylation signaling in the survival and maintenance of cerebellar Purkinje cells. They used a mouse model, in which the enzyme O-linked N-acetylglucosamine transferase (OGT) is selectively deleted in the cerebellum. Loss of OGT in Purkinje cells led to cerebellar atrophy, degeneration of dendrites and spines, and mice exhibited ataxia, rigidity, and posture abnormalities. Moreover, the authors performed a proteome analysis to detect differential protein levels of control and OGT KO mice. They identified several differentially expressed proteins and performed follow-up experiments for autophagy, mitophagy, and reactive oxygen species (ROS) pathways. They conclude that increased mitophagy and ROS generation are the main factors that lead to the degeneration of OGT-deficient Purkinje cells.
Experiments of the study are well performed, described, and show significant results.
However, I think that there exists a major misinterpretation of the data in regard to the effect of O-GlcNAcylation on mitophagy:
· The authors conclude that O-GlcNAcylation is required to inhibit mitophagy and that phenotypes in their OGT mouse model are due to overactivated mitophagy (see Title of the manuscript). But according to my knowledge and published data, it is quite the opposite. Reduced mitophagy leads to the accumulation of defective mitochondria and the generation of ROS in OGT-deficient cells (Murakami et al., Cell Reports 2021). In fact, activation of mitophagy rescued this effect. The same mechanism of has been linked to Parkinson’s disease. Loss of mitophagy regulators PINK1 and Parkin causes accumulation of mitochondria and neurodegeneration (Ashrafi and Schwarz, Cell Death Differ. 2013).
· The increase in PINK1 expression (Figure 4H) is most likely due to a compensatory mechanism of the cell to promote mitophagy. Overexpression of PINK1 was shown to rescue OGT deficiency phenotypes (Murakami et al., Cell Reports 2021).
· The accumulation of LC3 puncta in cells (Figure 5) could also indicate the accumulation of undegraded material and a defective autophagy process or might be independent of mitochondria.
· Figure 6A-C shows a small rescuing effect of Midiv-1 on Purkinje cell degeneration. However, the exact mechanism by which this rescue is achieved is not known. Mdivi-1 modulates mitochondrial function and oxidative stress also independently of mitochondrial fission inhibition (Ruiz et al., Front. Mol. Neurosci. 2018). Therefore, these data are not sufficient to draw a conclusion towards a link to mitophagy.
I strongly recommend reconsidering the conclusions drawn in this manuscript. And I suggest quantifying mitochondrial mass as shown previously (Murakami et al., Cell Reports 2021). This will provide insight into activated (decreased mass) or inhibited (increased mass) mitophagy function.
Author Response
Reviewer 1:
Experiments of the study are well performed, described, and show significant results.
However, I think that there exists a major misinterpretation of the data in regard to the effect of O-GlcNAcylation on mitophagy. The authors conclude that O-GlcNAcylation is required to inhibit mitophagy and that phenotypes in their OGT mouse model are due to overactivated mitophagy (see Title of the manuscript). But according to my knowledge and published data, it is quite the opposite. Reduced mitophagy leads to the accumulation of defective mitochondria and the generation of ROS in OGT deficient cells (Murakami et al., Cell Reports 2021). In fact, activation of mitophagy rescued this effect. The same mechanism of has been linked to Parkinson’s disease. Loss of mitophagy regulators PINK1 and Parkin causes accumulation of mitochondria and neurodegeneration (Ashrafi and Schwarz, Cell Death Differ. 2013). The increase in PINK1 expression (Figure 4H) is most likely due to a compensatory mechanism of the cell to promote mitophagy. Overexpression of PINK1 was shown to rescue OGT deficiency phenotypes (Murakami et al., Cell Reports 2021)· The accumulation of LC3 puncta in cells (Figure 5) could also indicate the accumulation of undegraded material and a defective autophagy process or might be independent of mitochondria. Figure 6A-C shows a small rescuing effect of Midiv-1 on Purkinje cell degeneration. However, the exact mechanism by which this rescue is achieved is not known. Mdivi-1 modulates mitochondrial function and oxidative stress also independently of mitochondrial fission inhibition (Ruiz et al., Front. Mol. Neurosci. 2018).
Therefore, these data are not sufficient to draw a conclusion towards a link to mitophagy. I strongly recommend reconsidering the conclusions drawn in this manuscript. And I suggest quantifying mitochondrial mass as shown previously (Murakami et al., Cell Reports 2021). This will provide insight into activated (decreased mass) or inhibited (increased mass) mitophagy function.
-We thank the reviewer for the valuable comments and suggestions.
As the reviewer mentioned, OGT triggers mitophagy to promote the uptake of defective mitochondria and cell survival in hematopoietic stem cells (Murakami et al., Cell Reports 2021). In this study, the result of quantitative immunostaining of TOMM20 showed that the mitochondrial mass was accumulated in OGT-deficient HSCs compared to control cells. However, the quantitative immunostaining of TOMM20 was decreased in OGT-deficient PCs (Figure S4) (page 7; lines 318-323). This result proves that mitochondrial mass was decreased in OGT-deficient PCs. Given that these results were observed from distinct cell types, the effect and mechanism of OGT deletion on mitochondrial damage in HSCs and PCs might be different.
In addition, mitophagy is an important mitochondrial quality control mechanism that selectively eliminates damaged mitochondria, but excessive activation of mitophagy can also lead to a decrease in the number of mitochondria and cell death (Yu Cao et al., EMBO Journal. 2023; Xing Guo et al., Nat Commun. 2016). And, for Midiv-1, a mitochondrial fission inhibitor, can modulate the mitochondrial functions and morphology, which is not a specific inhibitor of mitophagy. We agree that the role of up-regulated autophagy observed in OGT-deficient PCs is still unclear. Therefore, based on the reviewer’s suggestion and previous studies, we have turned down the claims and interpretation of the role of hyperactivated mitophagy in PC degeneration in the revised manuscript (page 15; lines 494-501). Thank you!
Reviewer 2 Report
The present manuscript by Liu et al. could be potentially intriguing and important, unfortunately this reviewer also raised many concerns on the lack of evidences and way of data presentation and their interpretation.
1) in Fig. 1, C, authors presented the data on calbindin-immunoreactivity. At P0, its immunoreactivity seemed to rather upregulated in OGT Knock out mice but they did not mention this. It is very important to explain this inconsistency.
2) The authors commented the dendrite and spine changes in their presentation. What was for axon in OGT knock out mice? It is essential to add such information in the text.
3) In relation of autophagy, authors othors only showed the data on LC3 expression for the judgement of autophgy. Authors need to present the expression of p62 protein and Beclin1 protein to understand full process of the autophagy in the cells.
4) Which is mitophagy or ROS production more important for PC neuron death? Moreover, it is essential to show the data on ROS production in MitoQ10-treated animals compared with control. Such data should be compared with the in-vivo effects on PC cell number in the control to verify that ROS production data are really comparable with the data on PC cell number in vivo.
5) There are some typographical and grammatical errors in the manuscript. The authors should seek English editing by native English speaker to amend the manuscript.
Author Response
Reviewer 2:
The present manuscript by Liu et al. could be potentially intriguing and important, unfortunately this reviewer also raised many concerns on the lack of evidences and way of data presentation and their interpretation.
-We thank the reviewer for the positive comments. All of the concerns raised by the reviewer have been carefully addressed. For details, please see the point-by-point responses listed below.
1) in Fig. 1, C, authors presented the data on calbindin immunoreactivity. At P0, its immunoreactivity seemed to rather upregulated in OGT Knock out mice but they did not mention this. It is very important to explain this inconsistency.
-We apologize for the misleading caused by the image in Fig. 1C. It should be noted that, as shown in Fig. 1D, we calculated the number of calbindin-positive cells, not the immunoreactivity of calbindin staining. To address the reviewer’s concerns, we have replaced the image of calbindin immunostaining of OGT cKO mice at P0 to avoid the misleading. Thank you for the nice suggestion!
2) The authors commented the dendrite and spine changes in their presentation. What was for axon in OGT knock out mice? It is essential to add such information in the text.
-Thank you for the great suggestion! The morphology of axons shows lots of discontinuous intumescence in OGT cKO mice, which has been shown in the Supplementary figure 2 (page 6; lines 274-277).
3) In relation of autophagy, authors only showed the data on LC3 expression for the judgement of autophagy. Authors need to present the expression of p62 protein and Beclin1 protein to understand full process of the autophagy in the cells.
-Thank you for your suggestion! Immunofluorescence analyses showed that loss of OGT in Purkinje cells dramatically down-regulated the expression of p62, whereas up-regulated the expression of Beclin1 (Supplementary figure 5). Together with the immunostaining results of anti-LC3 and LC3-GFP reporter mice (Fig. 5C and D), our results indicated that autophagy was dramatically up-regulated in Purkinje cells in OGT cKO mice (page 7; lines 332-339).
4) Which is mitophagy or ROS production more important for PC neuron death? Moreover, it is essential to show the data on ROS production in MitoQ10-treated animals compared with control. Such data should be compared with the in-vivo effects on PC cell number in the control to verify that ROS production data are really comparable with the data on PC cell number in vivo.
-Thank you for the great comments and suggestion! According to the results of MitoQ10 inhibitor treatment (Figure 6), ROS production seems more important for the death of PCs. The detailed results have been provided in the revised manuscript (Page 8; lines 352-356). Additionally, ROS detection results showed that MitoQ10 treatment can significantly inhibit the production of ROS (Supplementary figure 6) (page 8; lines 352-353). Together with the results showing in Figure 6, our results indicated that overproduction of ROS may lead to the reduced number of Purkinje cells.
5) There are some typographical and grammatical errors in the manuscript. The authors should seek English editing by native English speaker to amend the manuscript.
-Thank you for the nice suggestion! The revised manuscript has been carefully edited by English Language Editing service from MDPI's WebShop. And the typographical and grammatical errors have been corrected in the revised manuscript accordingly.
Round 2
Reviewer 1 Report
The authors have addressed my concerns regarding the effect of OGT deficiency on mitophagy function in the revised version of this manuscript. Conclusions on mitophagy activation would need to be built on at least two independent assays as Murakami et al. (Cell Reports 2021) show the opposite phenotype of OGT deficiency on mitophagy in their study.
It is possible that this opposite phenotype is due to cell type-specific differences, as mentioned by the authors in their response letter. Therefore, this discrepancy between the two studies shouldn’t be ignored but briefly discussed in the present study.
Author Response
Reviewer 1:
The authors have addressed my concerns regarding the effect of OGT deficiency on mitophagy function in the revised version of this manuscript. Conclusions on mitophagy activation would need to be built on at least two independent assays as Murakami et al. (Cell Reports 2021) show the opposite phenotype of OGT deficiency on mitophagy in their study. It is possible that this opposite phenotype is due to cell type-specific differences, as mentioned by the authors in their response letter. Therefore, this discrepancy between the two studies shouldn’t be ignored but briefly discussed in the present study.
- We thank the reviewer for the valuable comment and suggestion. We have added the information and discussion about the discrepancy of the role of OGT in PCs and HSCs in the revised discussion part of the manuscript (page 14, lines 471-486). Thank you!
Reviewer 2 Report
Although the authors tried to amend the manuscript, there are still criticisms to be coped with. The authors changed the title and put the significance of ROS production, but they did not describe how to measure ROS in the manuscript at all. This deficiency strongly weekens the manuscript. The revision is needed for the acceptance.
Author Response
Reviewer 2:
Although the authors tried to amend the manuscript, there are still criticisms to be coped with. The authors changed the title and put the significance of ROS production, but they did not describe how to measure ROS in the manuscript at all. This deficiency strongly weekens the manuscript. The revision is needed for the acceptance.
- We thank the reviewer for his/her great suggestion. The method of measurement of endogenous ROS level in PCs has been described in the revised “Materials and Methods” section of the manuscript (page 5, lines 197-206). Thank you!